# Monitoring the Establishment of VOC Gamma in Minas Gerais, Brazil: A Retrospective Epidemiological and Genomic Surveillance Study

**DOI:** 10.3390/v14122747

**Published:** 2022-12-09

**Authors:** Hugo José Alves, João Locke Ferreira de Araújo, Paula Luize Camargos Fonseca, Filipe Romero Rebello Moreira, Diego Menezes Bonfim, Daniel Costa Queiroz, Lucyene Miguita, Rafael Marques de Souza, Victor Emmanuel Viana Geddes, Walyson Coelho Costa, Jaqueline Silva de Oliveira, Eva Lídia Arcoverde Medeiros, Carolina Senra Alves de Souza, Juliana Wilke Saliba, André Luiz Menezes, Eneida Santos de Oliveira, Talita Emile Ribeiro Adelino, Natalia Rocha Guimaraes, Adriana Aparecida Ribeiro, Rennan Garcias Moreira, Danielle Alves Gomes Zauli, Joice do Prado Silva, Frederico Scott Varella Malta, Alessandro Clayton de Souza Ferreira, Ana Valesca Fernandes Gilson Silva, Poliane Alfenas-Zerbini, Flavia Oliveira de Souza, Adriano de Paula Sabino, Laura do Amaral Xavier, Natália Virtude Carobin, Alex Fiorini de Carvalho, Karine Lima Lourenço, Santuza Maria Ribeiro Teixeira, Ana Paula Salles Moura Fernandes, Flavio Guimarães da Fonseca, Jônatas Santos Abrahão, Felipe Campos de Melo Iani, Rodrigo Araújo Lima Rodrigues, Renan Pedra de Souza, Renato Santana Aguiar

**Affiliations:** 1Laboratório de Biologia Integrativa, Departamento de Genética, Ecologia e Evolução, Instituto de Ciências Biológicas, Universidade Federal de Minas Gerais, Belo Horizonte 31270-901, Brazil; 2Laboratório de Virologia Molecular, Departamento de Genética, Instituto de Biologia, Universidade Federal do Rio de Janeiro, Rio de Janeiro 21941-901, Brazil; 3Departamento de Patologia, Instituto de Ciências Biológicas, Universidade Federal de Minas Gerais, Belo Horizonte 31270-901, Brazil; 4Subsecretaria de Vigilância em Saúde, Secretaria de Estado de Saúde de Minas Gerais, Belo Horizonte 31585-200, Brazil; 5Pan American Health Organization-PAHO, Brasilia 70312-970, Brazil; 6Secretaria Municipal de Saúde, Prefeitura de Belo Horizonte, Belo Horizonte 30130-040, Brazil; 7Fundação Ezequiel Dias, Belo Horizonte 30510-010, Brazil; 8Centro de Laboratórios Multiusuários, Instituto de Ciências Biológicas, Universidade Federal de Minas Gerais, Belo Horizonte 31270-901, Brazil; 9Instituto Hermes Pardini, Belo Horizonte 30140-070, Brazil; 10Escola de Saúde Pública de Betim, Secretaria Municipal de Saúde, Prefeitura de Betim, Betim 32600-412, Brazil; 11Laboratório de Vírus, Departamento de Microbiologia, Instituto de Biotecnologia Aplicada à Agropecuária, Universidade Federal de Viçosa, Viçosa 36570-900, Brazil; 12Laboratório de Institucional de Pesquisa em Biomarcadores, LINBIO, Departamento de Análises Clínicas e Toxicológicas, Faculdade de Farmácia, Universidade Federal de Minas Gerais, Belo Horizonte 31270-901, Brazil; 13Centro de Tecnologia de Vacinas, Universidade Federal de Minas Gerais, Belo Horizonte 31310-260, Brazil; 14Laboratório de Vírus, Departamento de Microbiologia, Instituto de Ciências Biológicas, Universidade Federal de Minas Gerais, Belo Horizonte 31279-901, Brazil; 15Instituto D’Or de Pesquisa e Ensino (IDOR), Rio de Janeiro 22281-100, Brazil

**Keywords:** phylodinamics, PCR-genotyping, SARS-CoV-2 spread, genome sequencing, variants

## Abstract

Since its first identification in Brazil, the variant of concern (VOC) Gamma has been associated with increased infection and transmission rates, hospitalizations, and deaths. Minas Gerais (MG), the second-largest populated Brazilian state with more than 20 million inhabitants, observed a peak of cases and deaths in March–April 2021. We conducted a surveillance study in 1240 COVID-19-positive samples from 305 municipalities distributed across MG’s 28 Regional Health Units (RHU) between 1 March to 27 April 2021. The most common variant was the VOC Gamma (71.2%), followed by the variant of interest (VOI) zeta (12.4%) and VOC alpha (9.6%). Although the predominance of Gamma was found in most of the RHUs, clusters of Zeta and Alpha variants were observed. One Alpha-clustered RHU has a history of high human mobility from countries with Alpha predominance. Other less frequent lineages, such as P.4, P.5, and P.7, were also identified. With our genomic characterization approach, we estimated the introduction of Gamma on 7 January 2021, at RHU Belo Horizonte. Differences in mortality between the Zeta, Gamma and Alpha variants were not observed. We reinforce the importance of vaccination programs to prevent severe cases and deaths during transmission peaks.

## 1. Introduction

The emergence of a novel Betacoronavirus infecting humans in late 2019 led the world to this century’s most challenging global health crisis. The Coronavirus Disease 2019 (COVID-19) pandemic, caused by Severe Acute Respiratory Syndrome Coronavirus 2 (SARS-CoV-2), resulted in more than 621.87 million cases and 6.56 million deaths worldwide by October 2022 [1]. The pandemic caused collapses in health systems worldwide, especially in locations where effective measures to control viral spread were not practised or available [2].

Among the numerous lineages of SARS-CoV-2 described so far, some have called the attention of public health authorities, given their association with higher rates of viral transmission, disease severity, and resistance to the neutralizing antibodies elicited by prophylactic vaccines [3,4]. These lineages were defined as Variants of Concern (VOCs) and are responsible for increased disease incidence and transmission waves [1]. These VOCs were initially identified in the United Kingdom (Alpha), South Africa (Beta), Brazil (Gamma), and India (Delta) [5,6,7,8,9]. More recently, the VOC Omicron emerged with up to 46 new mutations in the SARS-CoV-2 genome (initially described in South Africa), which has spread rapidly, creating new COVID-19 waves, including in Brazil [10].

Since the beginning of the COVID-19 pandemic in Brazil, genomic surveillance studies showed the replacement of SARS-CoV-2 lineages over time [11,12,13]. Most of the early COVID-19 cases were caused by the introduction of B.1.1.28 and B.1.1.33 lineages, followed by the Zeta (P.2) variant in the southeast region of Brazil in July 2020, concurrent with the progressive decline of the number of cases. Despite the international introduction of VOC Alpha at the end of 2020, the number of cases associated with this variant never reached the same proportion in the UK or other countries in the northern hemisphere [14,15]. The VOC Gamma, first named P.1, emerged in Manaus, possibly in mid-November 2020, resulting in a new wave of infections and causing the local public health system to collapse [16]. Until 2020, the country had registered around 7.6 million cases and 195,000 deaths [17]. Four months later, another wave of infection occurred, reaching a 7-days moving average of 77,000 new cases and 3000 deaths per day [17]. The progression of the disease was observed in all states of the country. In Minas Gerais (MG), the fourth largest state in area and second most populous in the country, there was an increase of approximately 240% in the moving average of cases and 470% of deaths from January 01 to the peak of the disease in mid-April 2021 [17]. Brazil is the most populous South American country, borders ten different countries, and presents different socioeconomic groups that can favour the occurrence and emergence of VOCs and other lineages. Due to the size of the population, methods that aim to classify circulating VOCs and lineages that present low cost and quick results are essential for monitoring the spread of COVID-19.

Substantial decreases in COVID-19 cases and deaths were observed after approving safe and effective vaccines by the end of 2020 [18]. Since the beginning of vaccination programs, the number of severe cases and deaths have dramatically declined in Brazil. Until the beginning of April 2021, during VOC Gamma establishment, 13.6% population (28,696,332 doses) were vaccinated in all Brazilian territories, in which 1.1% (2294,840 doses) corresponded to the MG state population [17]. The emergence and dissemination of VOCs throughout Brazil highlighted the necessity and importance of genomic surveillance as a fundamental strategy in public health. Notwithstanding, previous studies suggested increasing the number of hospitalization and death cases with the emergence of VOC Gamma, but not at the patient level. Most studies compared the mortality rates in pre- and post-periods of Gamma introduction [2,4,7,19]. Herein, we conducted a retrospective epidemiological study in the MG State.

## 2. Materials and Methods

### 2.1. Study Design

We conducted a population-based retrospective study with samples from March 01 to 27 April 2021—Epidemiological Weeks (EW) 9-18/2021. MG is administratively divided into 28 Regional Health units (RHUs) (Appendix A). We conducted a two-stage sampling with 56% of the sample size allocated uniformly to all RHUs and 44% proportionally to each unit’s population density (Appendix A). Samples with *Ct* values under 30 were randomly selected (*n* = 1240). De-identified leftover nasopharyngeal samples were obtained from Secretaria do Estado de Saúde (SES-MG), a public organization responsible for COVID-19 diagnosis. Diagnostic tests were performed in subjects presenting flu-like or severe acute respiratory syndrome symptoms or hospitalized patients. The samples were processed in nine laboratories: *Universidade Federal de Minas Gerais* (*Laboratório de Biologia Integrativa, Laboratório de Vírus, Laboratório Institucional de Pesquisa em Biomarcadores, Núcleo de Ações e Pesquisa em Apoio Diagnóstico, CT Vacinas*), *Universidade Federal de Viçosa, Universidade Federal dos Vales do Jequitinhonha e Mucuri, Laboratório de Referência da Prefeitura Municipal de Belo Horizonte* and *Fundação Ezequiel Dias*. Research Ethics Committee approved the study (CAAE 33202820.7.1001.5348).

### 2.2. Epidemiological Data and Clinical Outcome Analysis

COVID-19 cases, deaths, and vaccination data were obtained from public databases https://coronavirus.saude.mg.gov.br/dadosabertos and https://saude.gov.br (Access date: 15 July 2022). Case and death heatmaps were generated. All analyses were performed in Rstudio version 4.2.3, using the packages Rlang [20], geobr [21], ggplot2 [22], ggaluvial [23], and sf [24]. Symptomatology and clinical outcome data were retrieved from Sistema de Informações de Vigilância Epidemiológica and e-SUS databases, respecting patient protection laws. Comorbidity, symptomatology, and clinical outcome data were available for 91,657, and 587 subjects, respectively. Logistic regression models were estimated on R.

To explore the transmissibility of SARS-CoV-2 lineages in MG, we investigated the RT-qPCR *Ct* values during the EW 45/2020–13/2021 (November 2020 to March 2021) (*n* = 119,507). Samples were diagnosed at the *Instituto Hermes Pardini* laboratory covering 343 (40.1%) municipalities in all the RHUs using the TaqPath COVID-19 CE-IVD RT-PCR kit (ThermoFisher Scientific, Burlington, ONT, Canada). The median for the genes MS2 (extraction control) and viral Nucleocapsid (N gene) were calculated per EW, and a curve was plotted using the R ggplot2 package [22].

### 2.3. cDNA Synthesis and rhAmp Genotyping Assay

The samples were genotyped using specific primers and probes to detect defining variant mutations (Figure 1). We synthesize cDNA using the High-Capacity cDNA Reverse Transcription Kit (Thermo Fisher Scientific) according to the manufacturer’s instructions. The cDNA was used as input in an RT-qPCR reaction using the rhAmp SNP Genotyping System technology (Integrated DNA Technology-IDT, San Diego, CA, USA) as described previously (Geddes et al., 2021). The concentration of each solution and final volume used for each genotyping reaction was: rhAmp Genotyping Master Mix (1X), rhAmp Reporter Mix (1.12X), and Specific SNP Primer (1.5X) in a final volume of 10 µL. Cycling conditions were 95 °C for 10 min and 60 cycles with 95 °C for 15 s, 57 °C for 1 min, and 68 °C for 30 s. We evaluated K417T (A22812C), E484K (G23012A), and N501Y (A23063T) mutations. The samples were classified as indicated in Figure 1. Samples with mutation profiles different from expected variants were denominated “others” and sequenced.

### 2.4. Library Preparation and Genome Sequencing

Sequencing was performed using two different technologies, Illumina (Illumina, San Diego, CA, USA) and IonTorrent (ThermoFisher Scientific, San Francisco, CA, USA), totaling 239 genomes. These genomes were derived from three different sampling groups. The first one consisted of 62 samples collected between 28 October 2020 and 28 February 2021. The second group consisted of 174 positive samples collected between 1 March 2021 and 31 March 2021. From this group, 34 samples were classified with the genotyping method as “other” lineages, and 140 samples were randomly selected for sequencing. The third group were composed of three samples with collection date after the genotyping sampling period (first week of May) (Figure 1).

The sequencing libraries were prepared using the QIAseq FX DNA Library Prep kit (QIAGEN, Germantown, MD, USA) and sequenced on the Illumina MiSeq platform (Illumina, San Diego, CA, USA) with v3 (600 cycles) cartridges, following all manufacturer’s instructions. Negative controls were used in each round of sample processing steps (cDNA synthesis, viral genome amplification, and library preparation). IonTorrent libraries were prepared using the Ion AmpliSeq SARS-CoV-2 Panel (ThermoFisher Scientific, San Francisco, CA, USA) and sequenced on the Ion Torrent PGM platform with a 314-chip kit (ThermoFisher Scientific, San Francisco, CA, USA) according to the manufacturer’s recommendations. Three negative controls were used in all sample processing steps (cDNA synthesis, viral genome amplification, and library preparation in each batch).

### 2.5. SARS-CoV-2 Genome Assembly and Lineage Classification

The sequencing data was processed following a previously described pipeline [25]. Briefly, raw reads were filtered with fastp v0.20.1 [26] to remove short and low-quality reads (Phred score < 30) and adapter sequences. The remaining reads were aligned against the SARS-CoV-2 reference genome (NC_045512.2) with Bowtie2 v2.4.2 [27]. Mapping files were then indexed and sorted with SAMtools v1.12 [28] and BCFtools v2.30.0 [28], and they were used to infer the consensus genome sequences. Finally, BEDtools [29] was used to mask low-depth sites (<10 × coverage). Sequences with less than 70% genome coverage were removed from the downstream analysis. The consensus sequences generated in our study were classified using the Pangolin tool v.3.1.14 and the NextClade web application v.1.7.0 [30]. Mutations associated with possible novel lineages have been manually verified in raw sequencing data. All consensus sequences were deposited on the GISAID EpiCOV database.

### 2.6. Phylogenetic and Phylodynamic Analysis of VOCs in Minas Gerais

A comprehensive reference dataset (*n* = 6110) was created with public genomes available on the GISAID EpiCOV database to confirm the lineage classification and contextualize the novel genomes. This dataset comprehends all Brazilian sequences, plus one international sequence per country per EW, including each country’s first reported SARS-CoV-2 genome (available on GISAID 11 April 2022). We also used as a reference the top ten sequences closely related to each genome generated in our study. The dataset was aligned with minimap v [31], and a maximum likelihood phylogeny was inferred with an IQ-tree v2.0.3 [32] under the GTR + F+I + G4 model [33,34].

To further contextualize the dynamics of the introduction and spread of SARS-CoV-2 VOC Gamma into MG, we assembled one additional dataset to perform the time-scaled phylogeographic reconstruction. Brazilian sequences from all states created the dataset. The number of sequences included is proportional to each federal unit (*n* = 1610, considering approximately 20 sequences for each of the 27 federative units of Brazil during January–May 2021). Given the size of the obtained datasets, an efficient maximum-likelihood-based method—Tree Time v [35]—was used to scale branch lengths in time and reconstruct phylogeographic history. The molecular clock analysis was performed using a fixed evolutionary rate (1 × 10^−3^), and a discrete symmetric model was used to estimate transitions among ancestral locations in the trees. The model comprehended six discrete categories, MG (*n* = 115) and five Brazilian politically defined regions: North (*n* = 376), Northeast (*n* = 583), CentreWest (*n* = 154), Southeast (except MG; *n* = 182), and South (*n* = 200). The tree was rooted in the divergence between P.1 and the clade P.1-like I [19]. Migration models were applied to infer the number of import and export cases to MG.

## 3. Results

### 3.1. Epidemiological Data and Clinical Aspects of SARS-CoV-2 in MG State

Here, we investigated the dynamics of cases and deaths in the MG state before introducing the VOC Gama. The first genome sequenced of VOC Gamma in MG was identified in EW 5 of 2021 (accessed on the GISAID EPICoV database on 30 January 2021). The number of cases and deaths was lower during the EWs 45–53 (2020) in contrast with increasing death rates observed by SES-MG during the first 13 EWs of 2021, after Gamma introduction (Figure 2A and Appendix A). In EW 45 (2020), 9637 positive cases and 189 deaths were confirmed. During the entire period evaluated in 2020, 189,630 cases and 3008 deaths were confirmed (for every 63 cases of COVID-19, one patient died). However, only in EW 8 (three weeks after Gamma’s first description in MG state), 37,578 cases and 837 deaths were confirmed (for every 44 cases of COVID, one patient died). The increasing rates in mortality were observed until the last EW evaluated (EW 13), in which 59,987 cases and 2174 deaths were confirmed (one died out of 28 cases). The epidemiological data suggests increasing rates of transmission and fatality after Gamma introduction in the state of MG (Figure 2A and Appendix A).

Gamma introduction in MG happened in a scenario where the vaccination process was starting only to the population classified as at-risk groups (healthy workers or individuals with ages superior to 80 years). Vaccination against COVID-19 in the MG state started during EW 4 (2021) and continued to increase during the consecutive weeks. In total, 2297,840 doses were distributed in the state during the entire period of our analysis (equivalent to almost 11% of the population of MG) (Figure 2B and Appendix A). Increasing transmission rates also followed the Gamma introduction in MG. An analysis from 119,507 RT-PCR-confirmed cases from 343 (40.1%) municipalities showed a significant decrease in *Cts* values from the N gene with no corresponding variation in endogenous cellular control (*MS2* gene) (Figure 2C). We observed a difference in medians between the EWs for the *Cts* referring to the N gene amplification (*p* < 0.0001; df = 4), and the EWs 5 and 13 were below the general median (considering all *Cts* from the EW evaluated in this study) (*Ct* = 18.34). The overall median *Ct* value was 25.78 (Appendix A). The higher viremia, represented by lower *Cts* values during Gamma epidemics, could explain the increasing transmission rates of this SARS-CoV-2 variant.

Moreover, our analysis showed an oscillation in the number of confirmed cases across the state during the time interval of this study (22 EWs) (Figure 3A). The RHU Belo Horizonte, which corresponds to the capital from MG, and presents a superior population density, maintained higher positivity rates, reaching 21,028 cases in the EW 12 after Gamma introduction. The RHU Uberlândia also showed increasing cases in the first weeks, reaching 5322 cases in EW 8. Nevertheless, there was a decrease in positive cases after the EW 10 (Figure 3A). At the end of the EW 17, the RHUs Belo Horizonte and Uberlândia had the highest positive cases in the state with 204,005 and 66,567 positive cases, respectively. RHU’s Januária and Pirapora (both in the North region) had a lower number of cases at the end of the study, corresponding to 6135 and 4643, respectively (Appendix A). The total number of cases recorded over time was reflected in the state’s mortality (Figure 3B). The RHU Belo Horizonte (Center region—Appendix A) maintained the higher numbers of deaths over the 17 weeks evaluated with a peak at EW 15, counting 610 deaths. The RHU Uberlândia had a rise in EW 10 (266 deaths) followed by a decrease, reaching 113 deaths in EW 17. The RHUs Januária and Pirapora also had the total number of cases reflected in mortality, representing 96 and 86 deaths, respectively (Appendix A). These results suggest a possible introduction of Gamma from the North-Triângulo region of MG state (RHU Uberlandia) coming from the north Brazilian area (Manaus, Amazon state), where Gamma was first described.

### 3.2. Circulation of SARS-CoV-2 Variants in MG State

We observed that the majority of the samples were classified as VOC Gamma (71.2%) (Figure 4A), followed by the VOI Zeta (12.4%) (Figure 4B). The VOC Alpha was also identified with a lower frequency (9.6%) (Figure 4C). Other genotypes presenting different combinations of lineage-defining mutations in the S gene (K417T, E484K and N501Y) were classified as “others” (6.8%) and targeted for whole genome sequencing (Figure 4D).

Heterogenous spatial variant distribution was seen. The VOI Zeta was present in 54.5% of the total samples from RHU Pedra Azul and 41.4% in RHU Ubá (Figure 4B and Appendix A). The VOC Alpha represented 53.2% of the total samples from RHU Coronel Fabriciano and 44.8% from RHU Teófilo Otoni (Figure 4C). “Other” lineages were particularly significant in the RHUs Pirapora and Alfenas—showing frequencies of 21.4% and 17.1%, respectively (Appendix A). A similar clustering pattern was found for VOC Gamma (Figure 4A). RHUs Patos de Minas and Uberlândia presented VOC Gamma in 100% of the samples, while RHUs Pedra Azul and Teófilo Otoni had the lowest relative frequencies at 30.3% and 24.1%, respectively.

For the samples classified as VOCs and VOIs, data related to symptoms, comorbidities, and clinical outcomes were obtained from SES-MG (Appendix A). The subjects participating in the study presented comorbidities such as asthma, heart disease, diabetes, immunosuppression, neurological disorders, obesity, pneumonia, and kidney disease (Figure 4E). Heart disease, diabetes, and obesity were most frequent in patients infected with the VOC Gamma (42.2%, 25.7%, and 10.8%, respectively). The most frequent comorbidities were heart diseases, diabetes, and obesity without association with SARS-CoV-2 variants (Appendix A).

The symptoms of cough, coryza, fever, headache, sore throat, dyspnea, anosmia, and ageusia were reported (Figure 4F). Fever, cough, and headache were the most reported symptoms presented by all patients independent of variants. No statistical differences were observed between Alpha, Gamma, Zeta, and other lineage symptoms, comorbidities and deaths excluding sore throat; Gamma-infected subjects were 40% less likely to present it than Zeta-infected patients (*p* = 0.034; OR: 0.60; 95%CI: 0.38–0.96) (Appendix A).

### 3.3. Sequencing Results

A subset of 239 novel SARS-CoV-2 genomes was generated. The average genome coverage was 97.84%, and the mean depth of 585x. Complete whole-genome sequencing metrics are available in Appendix A. According to the Pango lineages classification, 40% of the samples were identified as Gamma (36.4%; 87/239) or other five sublineages (P.1.1-1, P.1.7-1, P.1.14-10, P.1.15-15, P.1.17-1; total 28/239, 11.71%). Other lineages were also found: VOC Alpha (12.55%; 30/239), B.1.1.28 (3.35%; 8/239), VOI Zeta (26.78%; 64/239), P.4 (2.09%; 5/239), B.1.1 (1.67%; 4/239), B.1.1.33 (0.42%; 1/239), and P.7 (0.42%; 1/239) (Figure 5 and Appendix A).

### 3.4. Circulation of Other SARS-CoV-2 Lineages in MG

Eighty-four samples were classified as “other” lineages with different combinations of defining VOCs and VOIs spike mutations, from which 21 were sequenced. The Pangolin tool classified sixteen (76.19%) sequences as other lineages. One sequence (LBI412) in the phylogenetic analysis was close to the P.5 lineage clade as an ancestral group. However, these sequences do not present the same mutation profile as the P.5 reference (S: Q14K, T95I, G142D, N501Y, S640F, Q677H, D936G, V1176F), suggesting a novel genome lineage. Moreover, five sequences (LBI311, LBI312, LBI313, LBI401, and LBI406) were characterized as the P.4 lineage. The primary mutation that characterizes the P4 variant is the L452R and I720V in the spike gene. Another sample also had the mutation L452R but was classified as B.1.1 (LBI318 and LBI314). Our dataset also identified a sample classified as P.7 (LBI402) (Figure 5). Unlike the other variants identified in this study that have many mutations in the spike gene, the P.7 variant has only three: T21835C, A23403G/D614G, and G25088T/V1176F. Moreover, ten sequences were classified as B.1.1.28 (LBI315, LBI316, LBI321, LBI404, LBI405, LBI422, LBI423, and LBI424). Two genomes (LBI215 and LBI218) near the B.1.1.28 clade presenting the unique mutations E484Q and N501T forming a novel sub-cluster of B.1. In our phylogenetic analyses, five samples (LBI322, LBI325, LBI375, LBI378, and LBI379) classified as Gamma presented the unique E484K and N501T, being considered P.1-like genomes. All these results reinforce the importance of our strategy of genotyping/sequencing to screen large-scale samples increasing the chance of identifying novel SARS-CoV-2 genomes with higher frequencies of 2.5% in the population.

### 3.5. Phylodynamics of VOC Gamma Introduction into MG

A time-scaled phylogeographic reconstruction has been performed with a comprehensive dataset to explore the temporal and spatial dynamics of VOC Gamma introduction into the MG state (Figure 6). Analysis of the dataset estimated the origins of VOC Gamma on 19 October 2020 (90% CI: 7 September 2020–11 November 2020), with the earliest introduction in MG occurring by the beginning of 2021 (7 January 2021; 90% CI: 25 December 2020–25 January 2021). The model suggests more than 100 distinct introduction events in Brazil (103 events), culminating in diverse clades, revealing autochthonous transmission chains. The Southeast region is mainly responsible for importation cases in MG (with more than 70 introduction events), followed by the Northeast and South regions (each region with 14 and 8 introduction events). These events led to 16 different clades from MG, while others are represented by single sequences (singletons) (Figure 6). The most recent common ancestor for MG clades was 21 June 2020. Our time-scaled phylogeographic reconstructions coincide with the first confirmed infection by the Gamma variant in MG identified in mid-January 2021. Since then, the number of cases identified as the Gamma variant has been increasing, reaching the highest in March and April. Following the dominance of Gamma VOC in the MG state, new genomes were identified with lower frequencies (13.71%; 17/124) presenting new signatures such as K417T, synA23380T, and A845S mutations in the spike gene, suggesting diversification of this variant.

## 4. Discussion

Since the first identification of VOC Gamma, studies have reported the spread of this variant associated with increasing cases of and deaths from COVID-19, leading to a public health collapse in the country [2,25]. Our time-scaled phylogenetic inference suggests that VOC Gamma emerged in October–November 2020 in the Brazilian North region, as described by previous studies [7,36]. This variant became the most prevalent in all Brazilian states during the first months of 2021 and was responsible for Brazil’s second wave of COVID-19 [2]. COVID-19 cases and deaths also increased after VOC Gamma introduction in MG. The first introduction event was in the RHU Belo Horizonte (90% CI 7 January 2021; 90% CI: 25 December 2020–25 January 2021) and after that, through multiple locations, the Southeast region being responsible for most of the introduction events [5]. Several studies suggest that Gamma is more transmissible than other lineages, such as the previous Zeta and B.1.1.28, with higher viral loads on human samples [2,14,25]. Our study observed a reduction in *Ct* value for the Nucleoprotein (N) gene, corroborating the previous studies indicating that Gamma presents higher viral loads than the other lineages circulating in the same period in the MG state.

Higher viral loads (lower *Cts* values) were also observed, potentializing the virus transmission rates. Our results suggest a change in the epidemiological scenario after describing the first VOC Gamma genome in MG (30 January 2021), just one month after its identification in Manaus [5,7]. At the end of our study, the prevalence of Gamma cases was 71.2%. Gamma reached 100% prevalence in MG’s capital, Belo Horizonte, at the end of April 2021 [37]. Studies involving Gamma showed that this variant presents a higher spread rate and chances of developing severe cases and deaths related to different mutations in the spike gene [7,25]. The N501Y, first described in VOC Alpha, increases the affinity of spike viral protein to the human ACE2 receptor [9], with an additional effect when the K417T is present [16]. This process can contribute to the patient’s viral load and severe clinical outcomes [7].

COVID-19 symptom heterogeneity has been reported throughout the pandemic [38,39,40]. Many of the described comorbidities associated with severe COVID-19 were evaluated in our study. Patients with heart disease, diabetes, and hypertension comprise a large part of the risk groups for severe COVID-19 [40,41,42]. Heart diseases were reported in almost half of these patients, reinforcing the risk of cardiac comorbidities to develop severe cases of COVID-19. Patients diagnosed with diabetes were also frequent in our cohort. Chronic exposure to an abnormal metabolic environment can lead to disturbances in innate and adaptive immunity, cytokine storm release, and an increase in viral infection [41,42]. Together, all these factors can increase the risk of a severe prognosis for COVID-19 [39,41].

The study period coincides with the beginning of vaccination in Brazil. Official data indicates that 76% of the target population had already received the first dose, and 24% had received the second dose until the first week of April 2021. The available vaccines were Oxford-AstraZeneca (ChAdOx1) and CoronaVac (Sinovac Biotech, Haidian District, Beijing, China), responsible for more than 80% of applications at that time. Although the vaccination program was expected to reduce cases and deaths and improve patient prognosis [4,43], the vaccination strategy did not prevent the spread of Gamma during March and April 2021.

To corroborate the results found in our PCR-genotyping strategy, we selected 174 samples to further whole genome sequence analysis. Sequencing and phylogenetic analyses confirmed all PCR-genotyping results. Other lineages were also found in circulation with a much lower frequency. For example, the P.4 lineage, described as the second most prevalent in some regions of SP bordering MG, was found with low frequency. This lineage has 15 amino-acid mutations (six occurring in the spike gene, such as L452R and I720V). After its description in Porto Ferreira city, it was described circulating in 29 other cities within SP state (southeast region) [44]. Another case is the P.5 lineage, described for the first time in the states of SP and RJ. Only three samples of this lineage were found in our study [45]. P.5 presents 12 nonsynonymous mutations, of which five are in the Spike gene (F2L, Q14K, T95I, E484Q, and N501T). We also identified samples classified as P.7 lineage, first described in the SP state. Nevertheless, different from the P.7 reference sequence, our sequences harbour two additional mutations in the Spike gene (D614G and V1176F) and other positions on the genome N:P13L, ORF3a: T151I, and ORF9b P10S [46]. In the GISAID EpiCoV database (search on 28 September 2022), 271 sequences classified as P.4, 59 classified as P.5, and 282 genomes classified as P.7 are available. All P.4 and P.5 genomes available are from Brazil. More than 73% (208 genomes) of P.7 genomes were also identified in our country. These three lineages emerged in the first semester of 2021 in Brazilian states and descended from the lineage B.1.1.28 [44,45,46]. Those lineages were not found with high frequencies in other Brazilian regions; therefore, they are not considered a variant of interest or concern.

The first Brazilian identification of the VOC Alpha occurred in MG. However, Alpha frequencies were never significant statewide and were restricted to specific regions of the state. Those regions are related to intense human mobility to the USA as hotspots of immigration, where Alpha was dominant at the time of this study [47,48]. In this study, we also identified sub-lineages of Gamma, including P.1.1, P.1.14, P.1.15, and P.1.17. These sub-lineages can also be called P.1-like genomes and present additional mutations not identified in the first description of the VOC Gamma [25,49].

The Gamma dominance period was followed by VOC Delta in MG [50]. Delta’s introduction in Brazil and MG did not lead to the third wave of infection, later caused by the Omicron variant. It was first confirmed in MG in December 2021, reaching 100% of state infections in EW 4/2022. Omicron, its sub-variants (BA.2, BA.3, BA.4, and BA.5) and recombinants were responsible for most 2022 infections [51]. The higher transmission rates of Omicron associated with vaccine escape mutations and the relaxation of preventive measures may explain the increase in cases [52].

Our results reinforce the combination of genotyping and whole-genome sequencing as an effective strategy to improve and accelerate large-scale genome surveillance programs to detect and track the expansion of circulating lineages. With this strategy, more samples can be classified quickly, allowing real-time monitoring in limited resources scenarios. Our study presents limitations. We only used samples from symptomatic subjects, and it was impossible to retrieve information on symptoms and comorbidities presented by all individuals. Nevertheless, these results helped public health agencies monitor the spread of COVID-19 in MG. Moreover, genomic epidemiological surveillance must continue monitoring the state’s viral spread.

## 5. Conclusions

Overall, our study reports a genomic surveillance analysis based on PCR-genotyping and whole-genome sequencing that unveiled the epidemiological dynamics of the introduction of VOC Gamma in MG. In this study, we did not observe an association between virus load and the clinical outcomes evaluated for the VOC Alpha, Gamma, and VOI Zeta outcomes. Other lineages were also found circulating at low frequencies. From the phylogenetic reconstruction analysis, it was possible to establish that the southeast region was responsible for the state’s most significant number of introductions. Finally, the dispersion and dominance descriptions of the VOC Gamma were essential for understanding the evolutionary dynamics of SARS-CoV-2 contributing to epidemiological interventions.

## Figures and Tables

**Figure 1 viruses-14-02747-f001:**
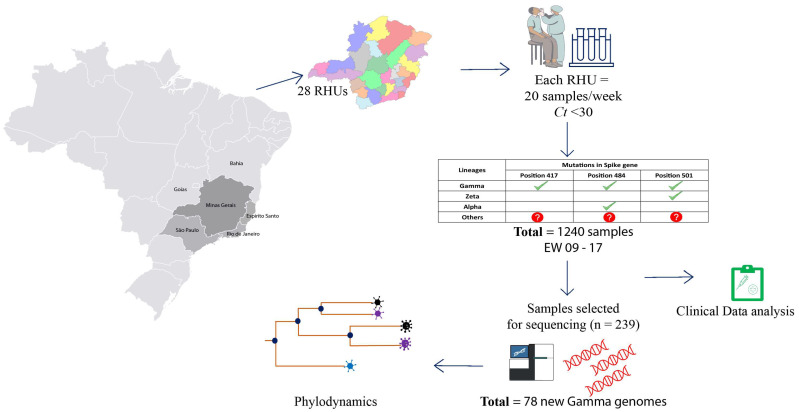
Workflow strategy for genomic evaluation of SARS-CoV-2 variants in MG. Twenty samples/week positive for SARS-CoV-2 with *Ct* < 30 were obtained from 28 Regional Health Units. Samples were screened for specific mutations in the Spike gene through PCR-genotyping (K417T, E484K and N501Y). Samples with mutations in each position are marked with green check symbol. Samples that presented different mutational profile are marked with red interrogation symbol. The description of the 1240 samples obtained, collection date, municipality of origin and genotyping result are included in Appendix A. A subset (239 samples) was selected for genome sequencing and phylogenetic analysis. A total of 78 new Gamma genomes were generated by our study and the genomes were used to infer the probable date of insertion in MG.

**Figure 2 viruses-14-02747-f002:**
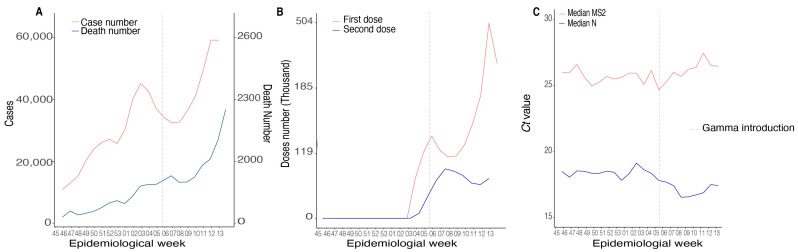
Epidemiological and viral data during Gamma introduction in MG state (45/2020–13/2021). (**A**) Numbers of positive and death cases in Brazil. (**B**) The first and second vaccine doses distributed in MG. (**C**) Distribution of median *Cts* values for SARS-CoV-2 *N* gene and *MS2* (intern control gene) from 119,507 positive samples from 343 municipalities from MG.

**Figure 3 viruses-14-02747-f003:**
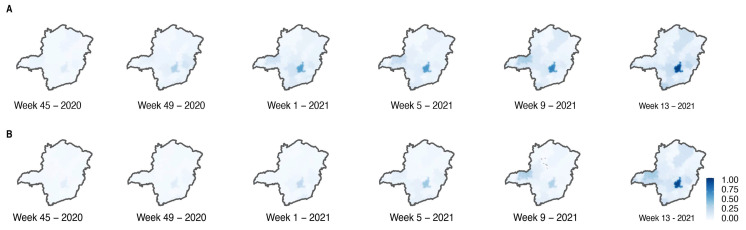
Spatio-temporal distribution of cases (**A**) and deaths (**B**) during Gamma introduction in MG.

**Figure 4 viruses-14-02747-f004:**
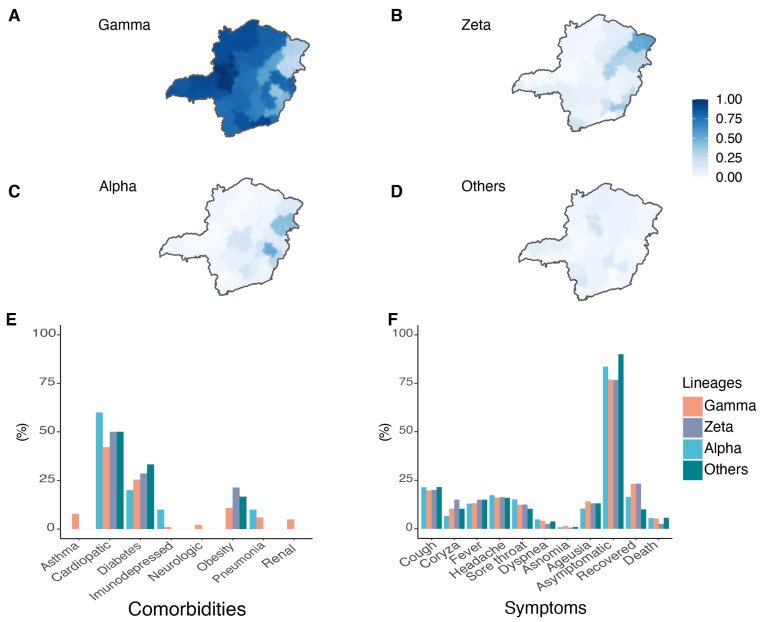
Distribution frequencies, symptoms and comorbidities of SARS-CoV-2 lineages in MG. Spatial frequency distribution of samples genotyped as VOC Gamma (**A**), Zeta lineage (**B**), VOC Alpha (**C**), and “Other” lineages (**D**). Comorbidities (**E**) and symptoms (**F**) reported by patients related to the SARS-CoV-2 variants identified.

**Figure 5 viruses-14-02747-f005:**
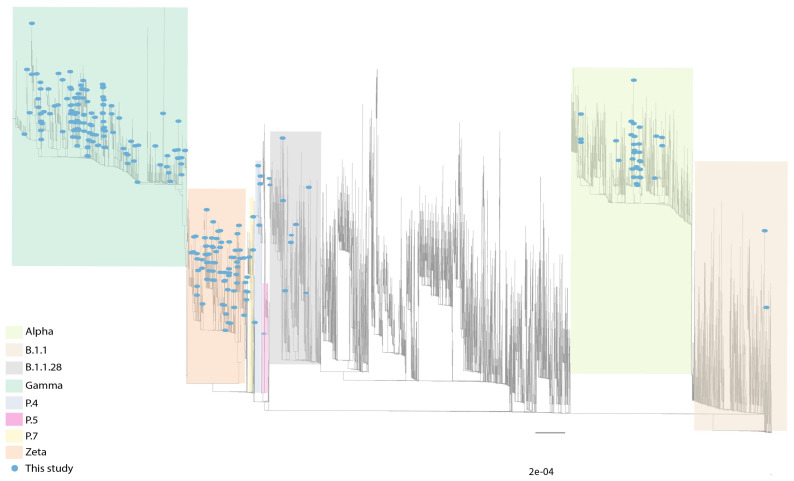
Maximum likelihood phylogeny inference corroborating the lineage classification using a global reference dataset (see material and methods section). Blue circles represent genomes generated in our study (n = 239). Grey lines represent reference genomes.

**Figure 6 viruses-14-02747-f006:**
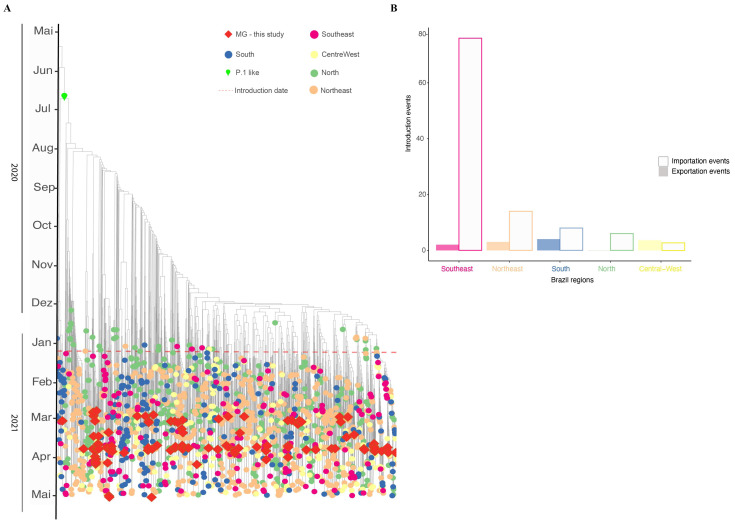
Timing and geographical introduction of novel Gamma genomes in MG state (molecular clock). (**A**) Time-scaled phylogeographic estimated with TreeTime under a fixed evolutionary rate (10^−3^) and a six-states symmetric discrete model (North, Northeast, Southeast, South, Centra-West and MG. (**B**) Number of SARS-CoV-2 introduction events for the MG state compared to other Brazilian regions, according to the timescale analysis.

## Data Availability

All consensus genome sequences characterized in this study have been deposited on GISAID and are publicly available (Appendix A).

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
