# Peer review of "Monitoring the Establishment of VOC Gamma in Minas Gerais, Brazil: A Retrospective Epidemiological and Genomic Surveillance Study"

_viruses, 2022, doi:10.3390/v14122747_

Round 1

Reviewer 1 Report

This is a well-prepared manuscript. The study aim is important and the methods applied are correct.

Please consider some minor changes that may improve the manuscript:

1. Please add 2-3 sentences on the COVID-19 pandemic in Brazil that may inform international leaders about the importance of genotyping in Brazil. E.g. Brazil is the most populous South American country, with a big population and different socioeconomic groups that may increase the risk of VOC occurrence. This will be important and interesting for international readers and underlines the importance of this manuscript.

2. Line 108 - please avoid double styles of citation "[17](OPAS., 2021)"

3. Please correct the ethical statement at the end of the manuscript.

4. Line 519 please add this page as a reference rather than a link inserted into the text.

5. Please clearly define the limitations of this study

6. Please add 2-3 sentences on practical implications and further research needs.

Author Response

This is a well-prepared manuscript. The study aim is important and the methods applied are correct.

Please consider some minor changes that may improve the manuscript:

  1. Please add 2-3 sentences on the COVID-19 pandemic in Brazil that may inform international leaders about the importance of genotyping in Brazil. E.g. Brazil is the most populous South American country, with a big population and different socioeconomic groups that may increase the risk of VOC occurrence. This will be important and interesting for international readers and underlines the importance of this manuscript.
  1. We appreciate the suggestion made by the reviewer. We added one sentence on Introduction (lines 106 - 110) and discussion (lines 729 - 734) about the importance of PCR-genotyping in Brazil.

  1. Line 108 - please avoid double styles of citation "[17](OPAS., 2021)"
  1. We changed the style of citation (line 153)
  1. Please correct the ethical statement at the end of the manuscript.
  1. We changed the ethical statement and add information about the supply of samples.
  1. Line 519 please add this page as a reference rather than a link inserted into the text.
  1. We changed the reference as requested by the reviewer.

  1. Please clearly define the limitations of this study
  1. We added two sentences explaining the limitations of our study (lines 746 - 749).
  1. Please add 2-3 sentences on practical implications and further research needs.
  1. We added two sentences at the end of our Discussion section (lines 749 - 752).

Reviewer 2 Report

In manuscript ID: viruses-2025328, entitled “Monitoring the establishment of VOC Gamma in Minas Gerais, Brazil: a retrospective epidemiological genomic surveillance study”, Alves and colleagues monitor the spread of SARS-CoV-2 variant Gamma in Minas Gerais, Brazil from March-April 2021.  They present SARS-COV-2 epidemiological distribution data along with PCR genotyping and sequencing results to identify the main variants and strains circulating in Minas Gerais during the study period.  They observed increased cases of COVID-19 and increasing death rates upon the introduction of VOC Gamma.  The authors conclude that although VOC Gamma was quickly fixed in the population in Minas Gerais, it did not lead to a third wave of infection.

Overall, the authors present a nice epidemiological study of SARS-CoV-2 infection in Minas Gerais, Brazil, from late 2020 to early 2021.  The strength of the study is the combination of clear epidemiological work with viral genotyping and whole genome sequencing.  The manuscript is clearly written except for a few minor problems listed below.  The results are supported by the data.  The methods are clearly described.  The figures provide a good visualization of the data, except for a couple of minor background issues listed below.  Once the issues listed below are addressed, my recommendation is for acceptance for publication.

Minor Problems:

1.    Section 2.1, Study Design:  Were the nasopharyngeal samples collected only from symptomatic people?  Or asymptomatic positive tests included?  Please add details to this method section. 

2.    The black background and color scheme in Figs. 1 and 6 make it difficult to interpret the figure.  Maybe a lighter background?  Some of the text within the images is clipped.

3.    Line 262-270:  What statistical test was used to determine a decrease in Cts following qPCR for the N gene following the introduction of VOC?  What was the median Ct from EWs 5-13?  What does the “general media” refer to and does it include EWs 5-13?

4.    Line 280:  Correct typo “pos-Gamma”

5.    Line 305:  Correct typo “MG satate”

6.    Line 424:  Correct typo “molecular cock”

Author Response

  1. Section 2.1, Study Design:  Were the nasopharyngeal samples collected only from symptomatic people?  Or asymptomatic positive tests included?  Please add details to this method section. 
  1. We appreciate the reviewer suggestion. The samples used in our study were provided by the Health Secretary of Minas Gerais State (SES-MG). All samples received had already been tested for COVID-19 and samples classified as positive were processed to our laboratory for genotyping and sequencing. The samples received come from individuals with flu or Severe Acute Respiratory Syndrome symptoms or hospitalized patients. This information was added in lines 161 - 164.

  1. The black background and color scheme in Figs. 1 and 6 make it difficult to interpret the figure.  Maybe a lighter background?  Some of the text within the images is clipped.
  1. We agree with the reviewer and changed the figures background in this new version of our manuscript.
  1. Line 262-270:  What statistical test was used to determine a decrease inCts following qPCR for the N gene following the introduction of VOC?  What was the median Ct from EWs 5-13?  What does the “general media” refer to and does it include EWs 5-13?
  1. From the Cts data, we calculated the median per epidemiological week, considering the epidemiological weeks 45-53/2020 and 1-13/2021 (see the method section 2.3). The median Ct for the EW 5 was 17.8 and EW13 was 17.41. The median Ct for 5-13 was 17.40. The general media refers to the media of all EWs evaluated in the study, including EWs 5-13. We added this information in lines 392 - 393.
  1. Line 280:  Correct typo “pos-Gamma”
  1. We changed the sentence as suggested.
  1. Line 305:  Correct typo “MG satate”
  1. We changed the sentence as suggested.
  1. Line 424:  Correct typo “molecular cock”
  1. We changed the sentence as suggested.